# 8-Hydroxydaidzein Downregulates JAK/STAT, MMP, Oxidative Phosphorylation, and PI3K/AKT Pathways in K562 Cells

**DOI:** 10.3390/biomedicines9121907

**Published:** 2021-12-14

**Authors:** Pei-Shan Wu, Chih-Yang Wang, Pin-Shern Chen, Jui-Hsiang Hung, Jui-Hung Yen, Ming-Jiuan Wu

**Affiliations:** 1Department of Applied Life Science and Health, Chia Nan University of Pharmacy and Science, Tainan 717, Taiwan; dc7575@gmail.com (P.-S.W.); pschen@mail.cnu.edu.tw (P.-S.C.); 2Ph.D. Program for Cancer Molecular Biology and Drug Discovery, Taipei Medical University, Taipei 11031, Taiwan; chihyang@tmu.edu.tw; 3Graduate Institute of Cancer Biology and Drug Discovery, Taipei Medical University, Taipei 11031, Taiwan; 4Department of Biotechnology, Chia Nan University of Pharmacy and Science, Tainan 717, Taiwan; hung86@mail.cnu.edu.tw; 5Department of Molecular Biology and Human Genetics, Tzu Chi University, Hualien 970, Taiwan; imyenjh@mail.tcu.edu.tw; 6Institute of Medical Sciences, Tzu Chi University, Hualien 970, Taiwan

**Keywords:** K562, 8-hydroxydaidzein, JAK/STAT, MMP, OXPHOS, AKT

## Abstract

A metabolite isolated from fermented soybean, 8-hydroxydaidzein (8-OHD, 7,8,4′-trihydroxyisoflavone, NSC-678112), is widely used in ethnopharmacological research due to its anti-proliferative and anti-inflammatory effects. We reported previously that 8-OHD provoked reactive oxygen species (ROS) overproduction, and induced autophagy, apoptosis, breakpoint cluster region-Abelson murine leukemia viral oncogene (BCR-ABL) degradation, and differentiation in K562 human chronic myeloid leukemia (CML) cells. However, how 8-OHD regulates metabolism, the extracellular matrix during invasion and metastasis, and survival signaling pathways in CML remains largely unexplored. High-throughput technologies have been widely used to discover the therapeutic targets and pathways of drugs. Bioinformatics analysis of 8-OHD-downregulated differentially expressed genes (DEGs) revealed that Janus kinase/signal transducer and activator of transcription (JAK/STAT), matrix metalloproteinases (MMPs), c-Myc, phosphoinositide 3-kinase (PI3K)/AKT, and oxidative phosphorylation (OXPHOS) metabolic pathways were significantly altered by 8-OHD treatment. Western blot analyses validated that 8-OHD significantly downregulated cytosolic JAK2 and the expression and phosphorylation of STAT3 dose- and time-dependently in K562 cells. Zymography and transwell assays also confirmed that K562-secreted MMP9 and invasion activities were dose-dependently inhibited by 8-OHD after 24 h of treatment. RT-qPCR analyses verified that 8-OHD repressed metastasis and OXPHOS-related genes. In combination with DisGeNET, it was found that 8-OHD’s downregulation of PI3K/AKT is crucial for controlling CML development. A STRING protein–protein interaction analysis further revealed that AKT and MYC are hub proteins for cancer progression. Western blotting revealed that AKT phosphorylation and nuclear MYC expression were significantly inhibited by 8-OHD. Collectively, this systematic investigation revealed that 8-OHD exerts anti-CML effects by downregulating JAK/STAT, PI3K/AKT, MMP, and OXPHOS pathways, and MYC expression. These results could shed new light on the development of 8-OHD for CML therapy.

## 1. Introduction

Leukemia is a hematologic disorder, which can be classified into four main types, including acute lymphocytic leukemia (ALL), chronic lymphocytic leukemia (CLL), acute myeloid leukemia (AML), and chronic myeloid leukemia (CML), according to the speed of progression and types of cells involved. Among these, CML is the most thoroughly studied. The cytogenetic hallmark of CML is the oncogenic fusion of *BCR**-ABL**1* (breakpoint cluster region-Abelson murine leukemia viral oncogene 1) resulting from translocation between chromosomes 9 and 22 [1]. The BCR-ABL1 fusion protein exhibits aberrant tyrosine kinase activity, which activates Ras/Raf/mitogen-activated protein kinase (MAPK) signaling pathways and leads to uncontrolled cell proliferation and promotes the malignant expansion of pluripotent stem cells [2]. Small-molecule tyrosine kinase inhibitors (TKIs) that block the adenosine triphosphate (ATP)-binding site of ABL1 represent the first- and second-line treatments of choice for most CML patients [3]. However, approximately 15~20% of patients fail to obtain optimal responses because of drug resistance or medication intolerance; therefore, non-TKI therapy represents a feasible adjunct treatment strategy for CML [4,5].

Janus kinase/signal transducer and activator of transcription (JAK/STAT) pathway participates in the development of many cancers [6]. JAK is a non-receptor protein tyrosine kinase and regulates BCR-ABL stability and oncogenic signaling in hematopoietic cells [7]. JAK activates STATs by phosphorylation, dimerization, and translocation into the nuclei, where STATs serve as transcription factors for hematopoiesis and leukemogenesis [8]. Emerging evidence has demonstrated that hyper-activation of STAT3 and STAT5 appears in primary leukemia and is correlated with poor survival rates. Therefore, there is an urgent need to develop either direct or indirect inhibitors of STAT3 and STAT5 [8,9]. Currently, STAT inhibitors that target upstream tyrosine kinases have yielded promising results. A decrease in phosphorylated STAT3 and an undetectable *BCR-AB1* transcript were observed after treatment with ruxolitinib (INCB18424), a JAK inhibitor, in combination with the TKI, nilotinib, in CML patients in a phase I trial (NCT01702064) [10]. The safety and efficacy of combining ruxolitinib with bosutinib, nilotinib, and dasatinib are being evaluated in a phase II trial (NCT03654768).

Matrix metalloproteinases (MMPs) are members of the protease superfamily of zinc-endopeptidases, which are responsible for degrading the extracellular matrix (ECM). They participate in multistep processes of invasion and metastasis and further promote tumor development [11]. Previous MMP inhibitors yielded unsatisfactory anti-tumor efficacies due to poor selectivity, specificity, or bioavailability. Recently, with advances in bioinformatics, proteomics, and drug delivery, small-molecule MMP-specific inhibitors and anti-MMP monoclonal antibodies are showing signs of resurrection in anti-cancer therapy [12].

The Warburg effect, that favors glycolysis over oxidative phosphorylation (OXPHOS) for energy production, was proposed as a universal metabolic alteration in carcinogenesis [13]. However, an increasing body of evidence suggests that both glycolysis and mitochondrial metabolism are upregulated in some cancers [14]. It was found that both glycolysis- and OXPHOS-related signaling pathways are activated in leukemia cells, such as THP-1, U937, KG-1, and K562, as well as cluster of differentiation 34 (CD34)-positive hematopoietic stem cells (HSCs), as a result of an enhanced need for energy metabolism [15]. It has become clear that TKI-resistant AML and CML, and leukemic stem cells (LSCs) are accompanied by increased OXPHOS, regardless of the presence or absence of genetic mutations; hence, interest has grown in developing clinically applicable OXPHOS inhibitors [16].

It is well-known that mitochondria play key roles in activating apoptosis in mammalian cells and B-cell lymphoma protein 2 (Bcl-2) family members are the key regulators thereof [17]. The Bcl-2 family consists of both anti-apoptotic, such as BCL2, BCL-xL and BCL-w, and pro-apoptotic members, such as BH3 Interacting Domain Death Agonist (BID) and BCL2 Associated X, Apoptosis Regulator (BAX) [17]. A selective BCL2 inhibitor, Venetoclax, can restore activation of apoptosis in leukemia [18,19]. Phase II clinical trials are ongoing to evaluate the efficacy of Venetoclax in combination with dasatinib (NCT02689440) and with cladribine, idarubicin, cytarabine (NCT02115295) in CML patients in the chronic or blast phase.

AKT, also known as protein kinase B (PKB), is a major downstream target of phosphoinositide 3-kinase (PI3K). AKT is activated via phosphorylation and translocation to nuclei and affects the activity of multiple transcriptional regulators which are involved in cell proliferation and survival [20]. The PI3K/AKT/mammalian target of rapamycin (mTOR) pathway plays a critical role in TKI-resistance in CML [21]. It was reported that NVP-BEZ235, a dual PI3K/mTOR inhibitor, effectively inhibited proliferation, and promoted apoptosis and autophagy in K562 cells [22].

*MYC* is a proto-oncogene and encodes a nuclear phosphoprotein that is involved in cell-cycle progression, apoptosis, and cellular transformation. It was proposed that ABL activates MYC through indirectly augmenting the acetylation of MYC and directly phosphorylating MYC [23]. Ubiquitination of MYC is considered an essential regulatory step in CML [24]. It was reported that a higher MYC expression level is associated with the progression of CML into a blastic crisis and poorer prognoses [25,26]. Recently, various types of potential direct and indirect MYC inhibitors have been developed to treat hematologic malignancies [25].

Isoflavones are dietary phytoestrogens mainly produced by the Fabaceae family [27]. They are used as chemical preventive agents in alternative therapy for cancers, metabolic syndrome, osteoporosis, and postmenopausal symptoms [28,29,30,31]. A hydroxylated derivative of daidzein, 8-hydroxydaidzein (8-OHD, 7,8,4′-trihydroxyisoflavone, NSC-678112), can be isolated from fermented soybean products. Recent research has also indicated the potential therapeutic effects of 8-OHD for inflammation, carcinogenesis, and melanogenesis [32,33,34,35,36,37]. Meanwhile, 8-OHD also had strong anti-proliferative activity in human promyelocytic leukemia HL-60 cells and K562 CML cells [38,39]. Similar to lots of natural compounds, 8-OHD exerts a hormetic effect in resistance to oxidative stress [40]. It scavenges free radicals at low doses; while produces reactive oxygen species (ROS) at high doses [32,33,37,39,41,42]. It may play dual roles, both antioxidant and pro-oxidant, in cancer prevention and treatment.

High-throughput technology uses an extensive holistic approach to detect differences in thousands of gene expressions from functional genomics and biological systems [41,42,43,44,45]. In this study, a microarray analysis was employed to investigate differentially expressed genes (DEGs) in 8-OHD-treated K562 cells, a model cell line derived from a female CML patient in blast crisis [46]. The gene ontology (GO) and Kyoto Encyclopedia of Genes and Genomes (KEGG) pathways were then analyzed [47,48,49]. In our previous study, we investigated the anti-CML effects of 8-OHD, focusing on cell-cycle regulation, cell apoptosis, autophagy, differentiation, and the altered level of BCR-ABL in K562 cells [39]. However, there are still many gaps in our knowledge of the roles of 8-OHD in CML, especially in terms of metabolism, the ECM during invasion and metastasis, and survival signaling pathways. Therefore, in the present study, we combined a bioinformatics analysis with experimental validation to investigate how 8-OHD regulates the JAK/STAT, MMP, and OXPHOS pathways. Furthermore, two hub genes, *AKT* and *MYC*, with high degrees of connectivity to biological process were found. These results shed new light on potential therapeutic targets for CML treatment.

## 2. Experimental Section

### 2.1. Preparation of 8-OHD (8-Hydroxydaidzein, 7,8,4′-Trihydoxyisoflavone, NSC-678112)

The isolation of 8-OHD from soybeans fermented with *Aspergillus oryzae* was conducted, from which the nuclear magnetic resonance (NMR) spectral data and purification were reported previously [33,34,37]. The chemical structure of 8-OHD is presented in Figure 1a.

### 2.2. Cell Culture

K562 cells were purchased from the Bioresource Collection and Research Center (Hsinchu, Taiwan) and cultured in RPMI-1640 medium supplemented with 10% fetal bovine serum (FBS), 1% nonessential amino acids (NEAAs), 100 units/mL of penicillin, and 100 µg/mL of streptomycin (Thermo Fisher Scientific, Rockford, IL, USA) in a 5% CO_2_ incubator at 37 °C.

### 2.3. Microarray Analysis

Cells were treated with vehicle (0.1% DMSO) or 8-OHD (50 and 100 μM) for 48 h. RNA was extracted from K562 cells with a Illustra RNA Spin Mini RNA Isolation Kit (GE Healthcare, Wauwatosa, WI, USA). A microarray analysis was performed as previously described [50,51]. All raw data were processed using the CLC Genomics Workbench v10.1 according to our previously established pipeline [52,53,54,55,56,57]. DEGs related to the treatment with 8-OHD were screened, and those with multiples of change values of >1.5 or <−1.5 and *p* < 0.05 were selected.

DAVID (v6.8) is a well-known web-based database with dominant gene functional classification, and various embedded biological process and pathway annotations. Gene ontology (GO) and KEGG analyses cluster targeted genes into different subgroups according to biological functions, signaling pathways, or diseases by an agglomeration algorithm method [52,53,54,55,56,57], and *p* < 0.05 was used as the cut-off criterion [58].

### 2.4. Pathway Enrichment Analysis

Molecular functions and the pathway database from the MetaCore platform (GeneGo, St. Joseph, MI, USA) were further used to explore potential signaling pathways and process networks modulated by 8-OHD. GO terms and a heatmap were used to summarize downregulated DEGs and pathways as we previously described [59,60,61].

DisGeNET (v6.0) is an integrating and standardizing knowledge management platform about disease associated genes and variants. It contains 628,685 gene-disease associations (GDAs), involving 17,549 genes and 24,166 diseases, and 210,498 variant-disease associations (VDAs), including 117,337 variants and 10,358 diseases [62]. 

The Search Tool for the Retrieval of Interacting Proteins (STRING) database contains a huge number of protein–protein networks from 5090 organisms, 24.6 million proteins, and more than 2000 million interactions [63]. In the present study, we used the STRING database (v11.5) to analyze protein-protein interacting (PPI) networks [63]. A Markov clustering algorithm (MCL) with inflation factor = 2 was employed to find cluster structure in graphs by a mathematical bootstrapping procedure [64].

### 2.5. Western Blot Analysis

Whole-cell lysates and nuclear extracts were prepared from cultured K562 cells respectively using RIPA lysis buffer and nuclear extraction kit (Cayman Chemical, Ann Arbor, MI, USA). Protein concentrations were measured by the Bradford assay (Bio-Rad Laboratories, Hercules, CA, USA).

Equal amounts of protein were loaded into the wells of 5~12% sodium dodecyl-sulfate polyacrylamide gel electrophoresis (SDS-PAGE). Following electrophoretic separation, proteins were further transferred to polyvinylidene difluoride (PVDF) membranes, which were then blocked with freshly made buffer (5% skim milk in phosphate-buffered saline (PBS) with 0.05% Tween 20, pH 7.4) and incubated with specific primary antibodies (Table 1) overnight at 4 °C. After rinsing, membranes were incubated with properly diluted horseradish peroxidase (HRP)-conjugated secondary antibodies (Jackson ImmunoResearch, West Grove, PA, USA) for 1 h. The signal on the membrane was developed by enhanced chemiluminescence detection (GE Healthcare).

### 2.6. RNA Extraction and Reverse Transcription-Quantitative Polymerase Chain Reaction (RT-qPCR)

RNA was extracted from K562 cells using an Illustra RNA Spin Mini RNA Isolation Kit (GE Healthcare). A High-Capacity cDNA Archive kit (Thermo Fisher Scientific) was used for cDNA synthesis. A qPCR was performed with Power SYBR Green PCR Master Mix (Thermo Fisher Scientific) in a 20 μL total volume that contained 0.4 μM of each primer (Table 2). The PCR program consisted of pre-incubation for 2 min at 95 °C, followed by 40 cycles at 94 °C for 15 s and 60 °C for 60 s (ABI StepOne Real-Time PCR System). A melting curve was performed to verify the amplification specificity. The relative mRNA expression was normalized to GAPDH expression and then calculated by the comparative Ct method. The primer pairs used in real-time PCR are given in Table 2.

### 2.7. Gelatin Zymography

K562 cells were treated with vehicle (0.1% DMSO) or 8-OHD (25, 50, and 100 μM) for 24 h, and media were collected by centrifugation to eliminate cells. Conditioned media was concentrated 10-fold by ultracentrifugation (Amicon Ultra-0.5, 3 kDa), and then analyzed by gelatin zymography in 10% PAGE containing 0.1% gelatin. After electrophoresis, gels were washed with 2% Triton X-100 to remove SDS. Gels were incubated in developing buffer (2% Triton X-100, 0.2 M NaCl, 0.05 M Tris-HCl, and 0.005 M CaCl_2_) overnight at 37 °C to induce gelatin lysis and then stained with staining solution (0.3% Coomassie blue, 20% methanol, and 7% acetic acid) for 4 h. Finally, gels were destained with water until the bands could clearly be seen.

### 2.8. Transwell Cell Invasion Assay

The in-vitro invasiveness of K562 cells was assayed using Matrigel^®^-coated Falcon 24-well transwell cell culture chambers (Boyden chambers) with 8-μm-pore-size polyethylene terephthalate filter inserts (BD Biosciences, Bedford, MA, USA). The chamber filters were coated with 50 mg/mL of Matrigel^®^ matrix overnight at 37 °C. K562 cells were treated with vehicle (0.1% DMSO) or 8-OHD (25 and 50 μM) for 24 h and collected by centrifugation to eliminate the medium. A 0.2-mL cell suspension containing 4 × 10^5^ treated cells in serum-free medium was added to the Matrigel^®^-coated filter inserts. Next, 750 μL of serum-containing medium was placed as a chemoattractant in 24-well plates. A light microscope was used to count invasive cells that had migrated into the lower chamber after K562 cells were cultured at 37 °C for 4 and 24 h [65,66].

### 2.9. Statistical Analysis

All experiments were repeated at least three times, and the values were expressed as the mean ± standard deviation (SD). Results were analyzed using a one-way analysis of variance (ANOVA) with Dunnett’s post-hoc test, and *p* < 0.05 was considered statistically significant.

## 3. Results and Discussion

### 3.1. Analysis of Microarray Data and Determination of DEGs

K562 cells were treated with vehicle (0.1% DMSO) or 8-OHD (50 and 100 μM) for 48 h, and transcript expressions were analyzed using the Human OneArray system, which contains 25765 known genes. We first performed sample-level quality control using a principal component analysis (PCA) and hierarchical clustering [67]. Figure 1b shows that PC1 and PC2 accounted for more than 85% of the data variance, and two duplicates of each treatment were closely clustered, with each treatment cleanly separated.

Next, the intensity data were pooled and calculated to identify DEGs based on the threshold of multiples of change of >1.5 or <−1.5, and *p* < 0.05. Correlations of genetic signatures between control samples and 8-OHD treatment conditions were analyzed by unsupervised hierarchical clustering. Figure 1c shows hierarchical clustering of GO terms, associated with DEGs of 8-OHD-treated samples (H1 and H5), revealing that H1 and H5 could be clustered into a group. The regulated genes for each GO term are listed in Appendix A. We intended to investigate the pathways and networks associated with 8-OHD-downregulated genes in this study; therefore, 1910 commonly downregulated DEGs were selected for further investigation (Figure 1d).

### 3.2. Enrichment of Downregulated Signaling Pathways Using MetaCore

We used the MetaCore bioinformatics suite to analyze pathways enriched by the 1910 downregulated DEGs, and the top 15 pathways are shown in Figure 2a. Downregulated genes for the top 50 pathways are listed in Appendix A. The JAK/STAT pathway is crucial in transferring extracellular signals to cells and initiating gene expressions involved in cell proliferation, differentiation, survival, and developmental processes [68,69]. STAT3 and STAT5A/STAT5B are of particular interest because they are downstream effectors of tyrosine kinase oncogenes and contribute to the development of hematopoietic malignancies through both canonical and non-canonical pathways [6,69]. We found that *STAT3* and *STAT5* appeared in nine out of the top 15 enriched pathways. These included “Development_Transcription regulation of granulocyte development”, “Prolactin/JAK2 signaling in breast cancer”, “IL-6 signaling in colorectal cancer”, “Cell cycle progression in prostate cancer”, “Regulation of microRNAs in colorectal cancer”, “Immune response_IL-4-induced regulators of cell growth, survival, differentiation and metabolism”, “Immune response_M-CSF-receptor signaling pathway”, “c-Myc in multiple myeloma” and “Development_Thrombopoietin signaling via JAK-STAT pathway”.

The MetaCore pathway map, which presents “Prolactin/JAK2 signaling in breast cancer” in response to 8-OHD treatment, is shown in Figure 2b. It was reported that STAT3 and STAT5 are constitutively activated in hematopoietic cancers [69]. Activated STATs induce transcription of pro-proliferative targets such as cyclin D1, CISH, SK4/IK1 and stimulate cancer cell proliferation [70,71,72]. Figure 2b,c show that *STAT3*, *STAT5A*, *STAT5B*, *c-Myb*, *PPIA* (*c**yclophilin A*), *CCND1* (*c**yclin D1*), *CISH*, and *KCNN4* (*SK4/IK1*) were significantly downregulated by 8-OHD. These results are consistent with our previous findings that JAK/STAT-mediated apoptosis/anti-apoptosis networks were significantly regulated by 100 μM 8-OHD in K562 cells [39].

Rajabi et al., 2020 also reported that 8-OHD triggered apoptosis in breast cancer stem-like cells through inhibiting the interleukin (IL)-6-mediated JAK2/STAT3 pathway [73]. STAT3 activation was found to be correlated with TKI resistance in CML [74]. Recently, transcriptome and reverse-phase protein arrays with a STRING analysis of differentially active proteins within TKI-persistent K562 cells revealed that STAT3 and AKT1 are central nodes [75]. We therefore further verified the involvement of the JAK2/STAT3 pathway in 8-OHD-treated K562 cells by Western blotting. Figure 3a shows that cytosolic JAK2 expression was dose- and time-dependently downregulated. Cytosolic STAT3 expression had profoundly decreased after 48 h in a dose-dependent manner and was correlated with transcriptomic data (Figure 2c). Furthermore, STAT3 phosphorylation was accordingly reduced (Figure 3a).

Phosphorylated (p)-STAT3 is translocated into nuclei and serves as a transcription factor in various gene expressions. Figure 3b shows that nuclear pan- and p-STAT3 dose-dependently decreased. This indicates that STAT3-regulated pathways were downregulated by 8-OHD in K562 cells.

### 3.3. 8-OHD Repressed MMPs Expression and Activities

It was reported that *BCR-ABL*^+^ cells secrete angiogenic factors, such as MMPs and vascular endothelial growth factor (VEGF), and stimulate angiogenesis [76]. Figure 4a shows a MetaCore pathway map of “Role of metalloproteases and heparanase in progression of pancreatic cancer” in response to 48 h of 8-OHD treatment. The most significantly downregulated genes, including *MMP14*, *MMP15*, *VEGFA*, *SOS1*, and *MMP11,* in the microarray analysis are shown as a heatmap in Figure 4b, and were reconfirmed by RT-q-PCR as shown in Figure 4c–g.

MMP14 (MT1-MMP) and MMP15 (MT2-MMP) are membrane-type (MT)-MMPs. Independent of proteolytic activity, these types of membrane-anchored MMPs are important modulators of cell-cell communication and intracellular signaling [77]. It was reported that *MMP14* is the target of BCR-ABL/ABL interactor 1 (Abi 1) signaling, and induces increased motility and invasiveness leukemic cells [78]. *MMP15* is highly expressed in AML patients and was correlated with poor overall survival [79]. Downregulation of *MMP14* and *MMP15* by 8-OHD may indicate that it possesses anti-metastatic activity (Figure 4c,d).

VEGFA promotes angiogenesis and is overexpressed in various tumors, including CML [80]. Drugs that target BCR-ABL, including imatinib, were reported to reduce *VEGFA* expression [81,82]. We found that 8-OHD, which could reduce BCR-ABL protein levels, also significantly downregulated *VEGFA* mRNA expression (Figure 4b,e).

Son of Sevenless 1 (SOS1), a guanine nucleotide exchange factor (GEF), is a dual activator of the small GTPases RAC and RAS. RAS activates MEK/ERK and PI3K/AKT pathways, which are essential for cell survival and proliferation [83]. The SOS1-RAC axis is critical for CML transformation and the leukemogenic potential of BCR-ABL [84]. Recently, it was found that *SOS1* is one of the essential genes which have higher expression and encode proteins that engage in PPIs in K562 cells [85]. Selective and potent small-molecule inhibitors of SOS1 were developed to block RAS activity in tumor cell lines [86]. We found herein that 8-OHD could dose-dependently downregulate *SOS1* transcription (Figure 4f).

Most MMPs are secreted as inactive forms and activated by cleavage of extracellular proteinases. On the other hand, MMP11 is processed intracellularly and secreted in its active form. MMP11 plays a vital role in the progression of epithelial malignancies, such as colon cancer, breast carcinoma, esophageal cancer, basal cell carcinoma, and others; but there are no reports of its role in CML progression [87]. We found that 8-OHD dose-dependently repressed *MMP11* expression. However, the detailed underlying mechanism by which 8-OHD downregulates the above gene expressions needs further study.

Most studies of myeloproliferative malignancies focused on expressions or activities of MMP9 and MMP2, which are secreted gelatinases [76,88,89,90]. Neither of these two MMPs was downregulated in the microarray assay; we therefore analyzed MMP activity in cultured medium by gelatin zymography. We found that K562-secreted MMP9 activity (96 kDa) could be dose-dependently inhibited by 8-OHD (Figure 5a).

We previously reported that 100 μM 8-OHD exerted about a 30% inhibitory effect on K562 proliferation after 24 h of treatment, while 25 and 50 μM, 8-OHD respectively, showed less than 10% and 20% inhibition [39]. To explore the effects of 8-OHD on cellular invasion, K562 cells were treated with the vehicle, or 25 or 50 μM 8-OHD for 24 h. Then treated cells were collected by centrifugation, washed and resuspended in serum-free medium. Equal amounts of cells were seeded in the Matrigel^®^-coated inserts while the lower chambers were filled with medium containing serum as a chemoattractant. Figure 5b shows that the invasive ability of 8-OHD-treated K562 cells through an ECM-coated membrane was significantly lower than that of the vehicle-treated cells, in a dose-dependent manner, in both the 4- and 24-h assays (*p* < 0.01). In conclusion, 8-OHD represses MMP gene expression and activity, and inhibits K562 cell invasion.

### 3.4. 8-OHD Repressed OXPHOS

It is known that enrichment of DEGs by the multiple of change may discard many useful genes. We thus re-selected target genes commonly dose-dependently downregulated by 50 and 100 μM of 8-OHD using normalized data. An entire set of 5042 downregulated genes was selected and subjected to functional annotation and clusterization using the DAVID bioinformatics tools. The top KEGG pathways are shown in Figure 6a. It was found that “metabolic pathways” were the most significantly associated, followed by “oxidative phosphorylation” and three neurodegenerative diseases: Parkinson’s, Huntington’s, and Alzheimer’s diseases. These results agree with our finding in Figure 1c that “mitochondria” was the top GO term enriched by 8-OHD-downregulated DEGs (*p* = 1.81 × 10^−33^) (Appendix A).

Using a global mRNA microarray analysis combined with an Ingenuity Pathway Analysis, Flis et al. found that CML stem cells display enhanced expression of OXPHOS genes [91]. Furthermore, disrupting the abnormal increase in OXPHOS in TKI-resistant CML stem cells can prevent disease relapse [92]. We previously reported that OXPHOS (GO:0006119) was significantly associated with 100 μM 8-OHD-downregulated DEGs [39]. Here, we further demonstrated the effects of 8-OHD on KEGG “oxidative phosphorylation” pathway as shown in Figure 6b. Expressions of most genes in complex I (NADH dehydrogenase) were repressed in response to 8-OHD. Expression patterns of some significantly downregulated genes are further shown in a heatmap in Figure 6c.

We further verified expressions of some genes by RT-qPCR. It was found that 8-OHD dose-dependently repressed four complex I genes, *NDUFB10* (NADH: ubiquinone oxidoreductase subunit B10), *NDUFV1* (NADH: ubiquinone oxidoreductase core subunit V1), *NDUFS7* (NADH: ubiquinone oxidoreductase core subunit S7), and *NDUFB8* (NADH: ubiquinone oxidoreductase subunit B8) after 48 h treatment (Figure 7a–d). Four others, *SDHB* (succinate dehydrogenase complex iron-sulfur subunit B, Complex II), *CYC1* (cytochrome C1, complex III), *COX5A* (cytochrome C oxidase subunit 5A, complex IV), and *ATP5D* (also known as *ATP5F1D*, ATP synthase F1 subunit δ, complex V), were also significantly downregulated by 8-OHD in a dose-dependent manner (Figure 7e–h). In conclusion, our result demonstrates that 8-OHD acts as an OXPHOS repressor, alters energy metabolism, and inhibits K562 proliferation.

We previously reported that 8-OHD causes caspase-7-dependent apoptosis, and mitochondrial integrity is under the control of Bcl-2 family members [39,93]. We thus further investigated how 8-OHD affects expressions of Bcl-2 family. Microarray data revealed that only pro-apoptotic *BID,* but not anti-apoptotic *BCL2,* was downregulated by 8-OHD (Appendix A). Western blotting analysis showed that pro-apoptotic BAX protein was not altered by 8-OHD treatment (Appendix A). These results indicate that 8-OHD induces apoptosis through a Bcl-2-independent pathway in K562 cells.

### 3.5. PI3K/AKT Pathway Is Resposible for K562 Cell Cycle Arrest

Genes associated with ML were collected from the DisGeNET database (http://www.disgenet.org/; accessed on 1 November 2021), which led to a collection of 385 pathological targets of ML (C0023470). Subsequently, the intersection of ML-associated genes and 8-OHD-downregulated 5042 genes was regarded as putative targets of 8-OHD against ML. As shown in Figure 8a, 112 genes were obtained in this manner. The top 15 KEGG pathways associated with these 112 targeted genes are shown in Figure 8b. Detailed information and downregulated genes involved in the top 25 KEGG pathways are shown in Appendix A. “Chronic myeloid leukemia” (hsa05220) and “PI3K-Akt signaling” (hsa04151) are the top 2 and 4 on the list and the 8-OHD-downregulated genes involved in these two pathways are indicated by red marks in Figure 8c,d. It was found that PI3K/AKT signaling and BCR-ABL-induced gene expressions of the *SOS-RAC* axis, *STAT5*, and *MYC* were downregulated by 8-OHD in the CML pathway (hsa05220) (Figure 8c). We previously reported that 8-OHD caused K562 cell-cycle arrest at the S phase by inhibiting p21Cip1 expression and upregulating CDK6 and cyclin D2 expressions [39]. The current data further demonstrated that the PI3K/AKT pathway is responsible for the upstream signaling of 8-OHD-induced cell cycle arrest (Figure 8d).

### 3.6. AKT and MYC Are Hub Proteins Downregulated by 8-OHD

We then used the STRING database to analyze potential PPI networks for the 112 gene targets. Markov clustering algorithm (MCL) was selected to classify proteins into different clusters; the resulting network was divided into six clusters that contained most of the genes associated with cancer progression and metastasis (Figure 9a). The proteins enriched in AML and CML are respectively shown in blue and red. There were nine common proteins, including AKT1, MYC, NRAS, KRAS, PIK3CB, SOS1, MAPK3, STAT5A, and STAT5B. It clearly showed that AKT1 and MYC are two hub proteins, with 45 and 46 edges, respectively. A Western blot analysis of AKT activation and MYC expression was employed to reconfirm the bioinformatics data. Figure 9b shows that phosphorylation of AKT was pronouncedly inhibited by 8-OHD in a time- and dose-dependent manner. It was reported that inhibitors of PI3K/AKT/mTOR pathway can enhance CML apoptosis and autophagy and increase CML sensitivity to TKIs [22]. In combination with our previous finding that 8-OHD can induce apoptosis and autophagy [39], the current study further demonstrated that these effects are related to downregulation of the PI3K/AKT/mTOR pathway.

Figure 9c shows that the nuclear c-Myc level was decreased by 8-OHD. The inhibition profile was less effective than that of AKT and was completely correlated with that of BCR-ABL in our previous report [39]. This result supports 8-OHD being able to repress both ABL and MYC oncogene expressions and inhibit uncontrolled CML proliferation.

## 4. Conclusions

The bioinformatics data revealed that 8-OHD downregulated JAK/STAT, PI3K/AKT, MMP, and OXPHOS pathways in K562 cells. Validated data from Western blotting revealed that cytosolic JAK2 expression and phosphorylation of STAT3 decreased in dose-dependent manners in 8-OHD-treated K562 cells. Meanwhile, the repression of *MMP14*, *MMP15*, *VEGFA*, *SOS1*, and *MMP11* by 8-OHD may indicate that it possesses anti-metastatic activity. Gelatin zymography and the transwell assay further confirmed that K562-secreted MMP-9 and invasive activities were inhibited by 8-OHD. In addition, RT-qPCR data revealed that 8-OHD dose-dependently repressed OXPHOS- and energy metabolism-related genes, including *NDUFB10, NDUFV1*, *NDUFS7*, *NDUFB8*, *SDHB*, *CYC1*, *COX5A*, and *ATP5D*. In combination with DisGeNET, we found that 8-OHD-downregulated PI3K/AKT is a key pathway for CML development. The PPI network also indicated that AKT1 and MYC are two hub proteins for cancer progression and metastasis, and the Western blot analysis of AKT activation and MYC expression was employed to reconfirm these prediction data. Collectively, the combination of bioinformatics analyses and experimental validation has shed new light on the development of 8-OHD for CML therapy.

## Figures and Tables

**Figure 1 biomedicines-09-01907-f001:**
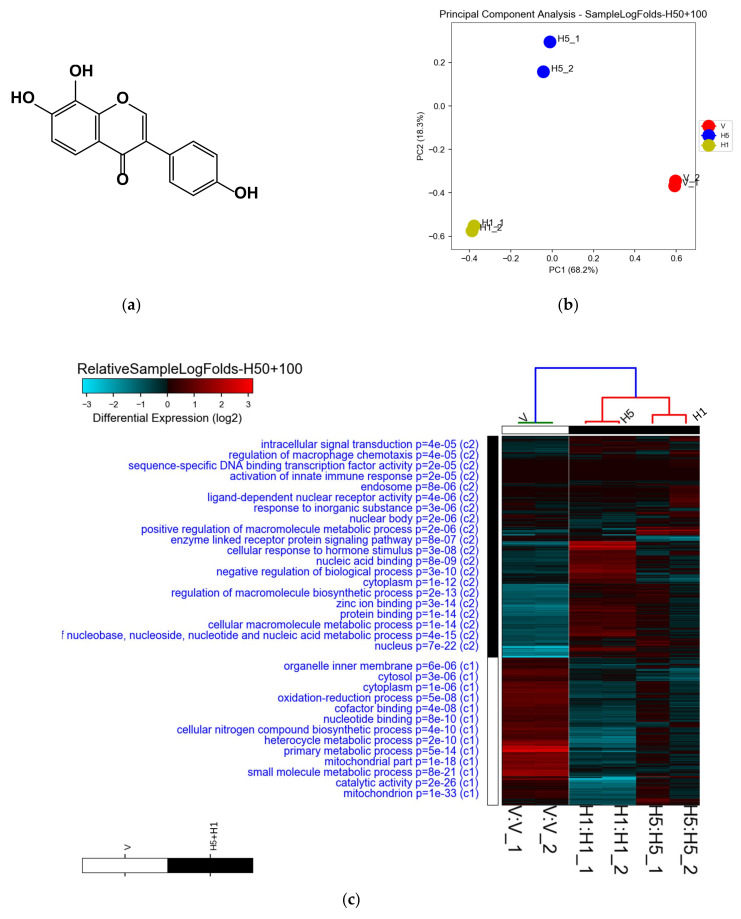
Microarray analysis of 8-OHD-treated K562 cells. (**a**) Chemical structure of 8-OHD. (**b**) Principal component analysis (PCA) of microarray data obtained from analyzing duplicate samples of vehicle (V) and 8-OHD treated at 50 μM (H5) and 100 μM (H1). (**c**) Heatmap of enriched gene ontology (GO) terms associated with 50 and 100 μM 8-OHD differentially expressed genes (DEGs). GO processes listed in the upper half are terms of upregulated DEGs. The lower processes are GO terms of downregulated DEGs. Only some of the representative GO terms are shown. (**d**) Venn diagrams demonstrating the numbers of common downregulated DEG between those treated with 50 μM 8-OHD (H5) and 100 μM 8-OHD (H1) compared to that of vehicle (V), based on the threshold of multiples of change of >1.5 or <−1.5, and *p* < 0.05. The '^' symbol represents the intersection of two sets.

**Figure 2 biomedicines-09-01907-f002:**
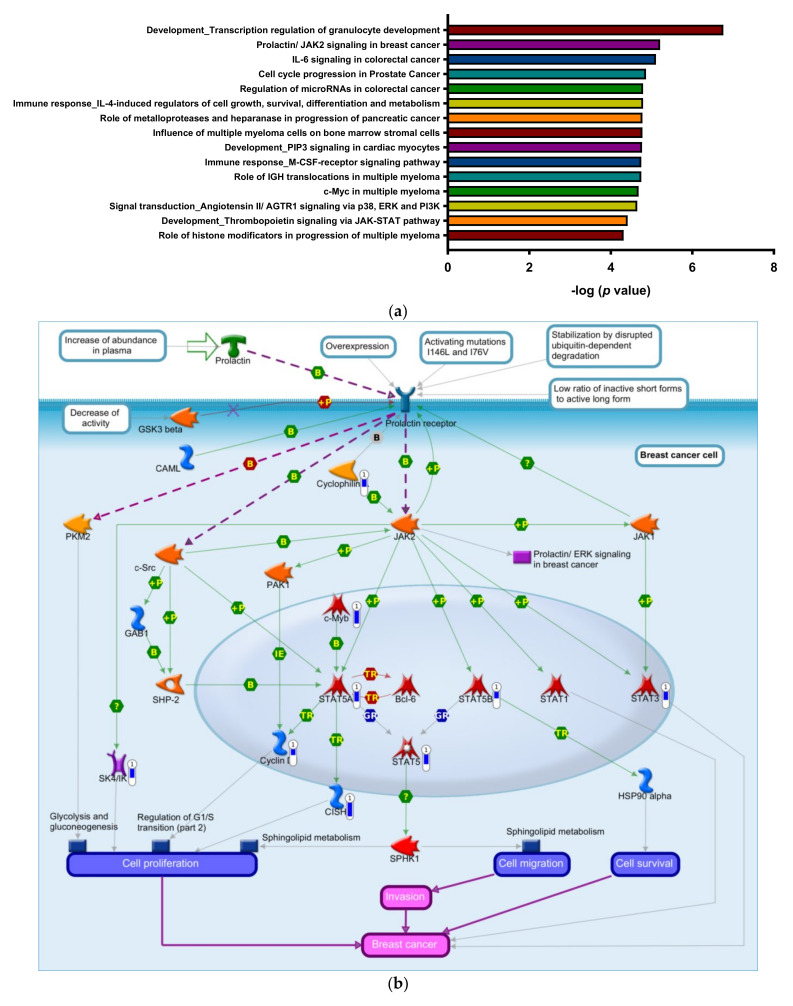
MetaCore pathways downregulated by 8-OHD treatment in K562 cells. (**a**) The top 15 significantly enriched pathways according to the MetaCore bioinformatics suite in response to 8-OHD treatment. (**b**) The MetaCore pathway map “Prolacin JAK2 signaling in breast cancer” was obtained by functional analysis of 1910 downregulated differentially expressed genes (DEGs) in response to 8-OHD. The blue thermometer indicates downregulated genes, and the thermometer level represents the intensity of the log_2_(fold change). Image generated using MetaCore. (**c**) Heatmap of expression patterns of Prolacin JAK2 signaling-associated genes in response to 8-OHD. H5 and H1 represent 50 and 100 μM 8-OHD, respectively.

**Figure 3 biomedicines-09-01907-f003:**
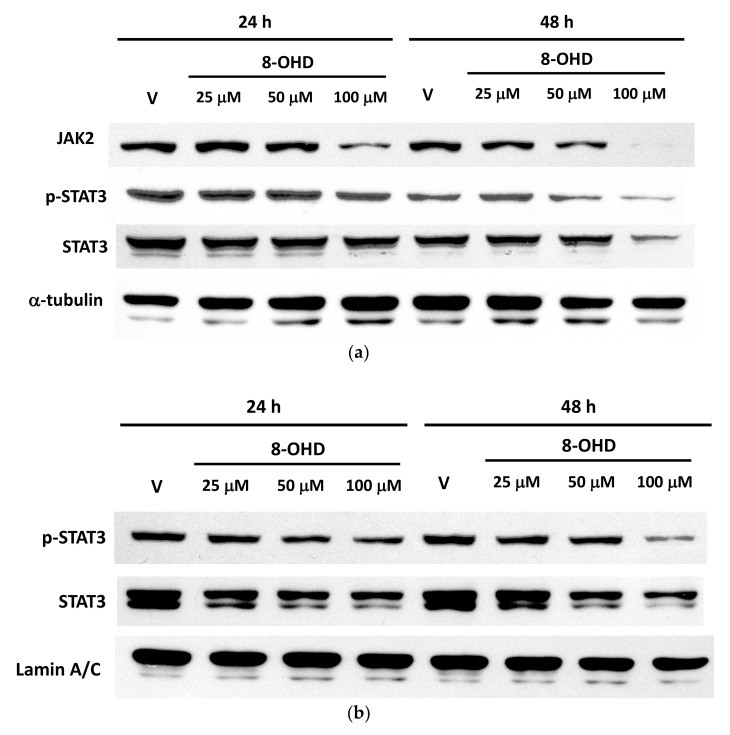
Western blot analysis of the Janus kinase (JAK)/signal transducer and activator of transcription 3 (STAT3) signaling pathway in 8-OHD-treated K562 cells. (**a**) Cell lysates were prepared after 24 and 48 h of treatment. Equal amounts of protein lysates were used to examine the expressions of JAK2, STAT3, and phosphorylated (p)-STAT3 by a Western blot analysis. α-Tubulin levels were used as a control to ensure that equal amounts of protein were present. (**b**) Nuclear extracts were prepared after 24 and 48 h treatment. Equal amounts of nuclear lysates were used to examine the nuclear levels of pan- and p-STAT3, and lamin A/C levels were used as a loading control.

**Figure 4 biomedicines-09-01907-f004:**
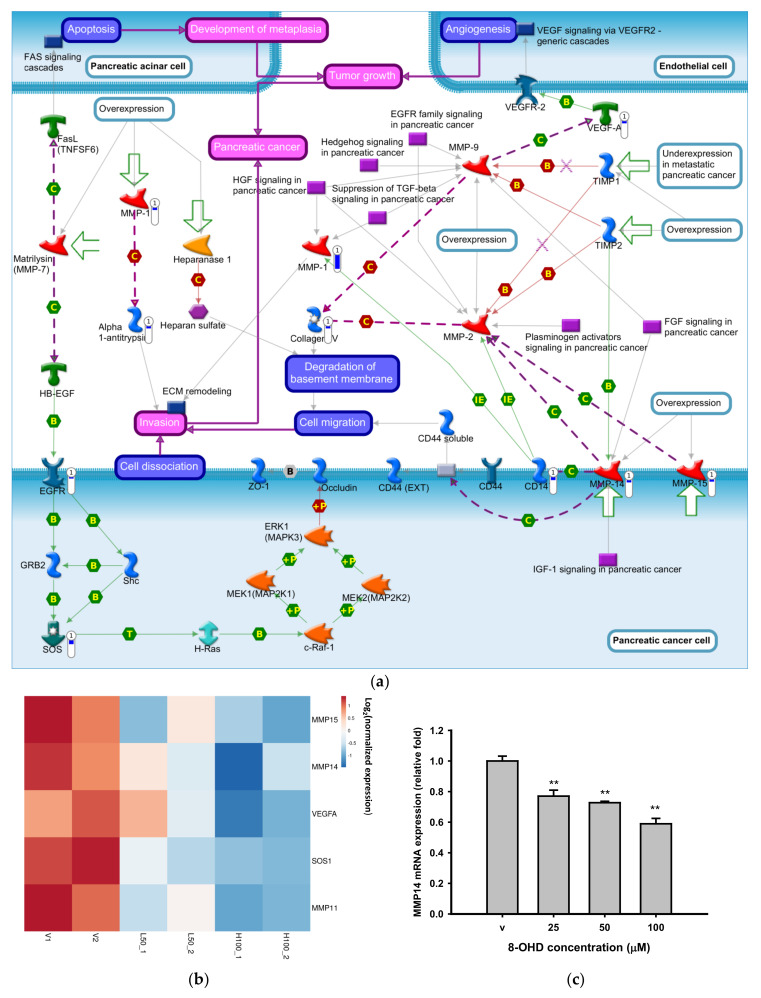
8-OHD repressed matrix metalloproteinase (MMP) expressions. (**a**) The MetaCore pathway map of the “Role of metalloproteases and heparanase in progression of pancreatic cancer” obtained by functional analysis of 1910 downregulated differentially expressed genes (DEGs) in response to 8-OHD. The blue thermometer indicates downregulated genes, and the thermometer level represents the intensity of the log_2_(fold change). The image was generated using MetaCore. (**b**) Heatmap of expression patterns of MMP-associated genes in response to 8-OHD. H5 and H1 represent 50 and 100 μM 8-OHD, respectively. (**c**–**g**). Reconfirmation of MMP-associated genes by RT-qPCR. * *p* < 0.05 and ** *p* < 0.01 indicate significant differences compared to vehicle-treated cells.

**Figure 5 biomedicines-09-01907-f005:**
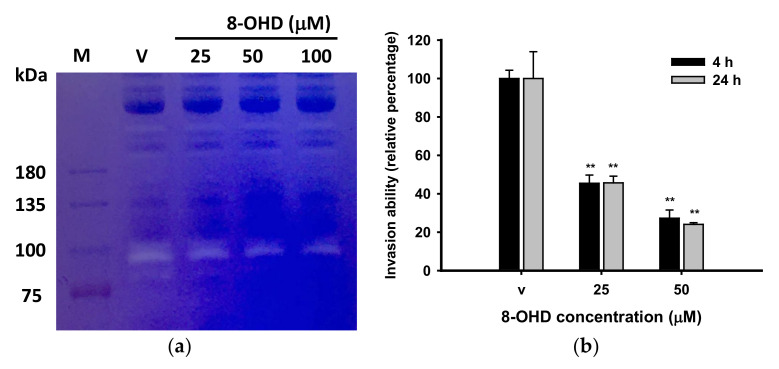
8-OHD repressed K562 matrix metalloproteinase (MMP) activity and invasion ability. (**a**) Gelatin zymography for the determination of secreted MMP9 activity in K562 cells treated with 8-OHD for 24 h. (**b**) Cell invasion was tested using a modified Boyden chamber assay as described in Section 2. ** *p* < 0.01 indicates a significant difference compared to vehicle-treated cells.

**Figure 6 biomedicines-09-01907-f006:**
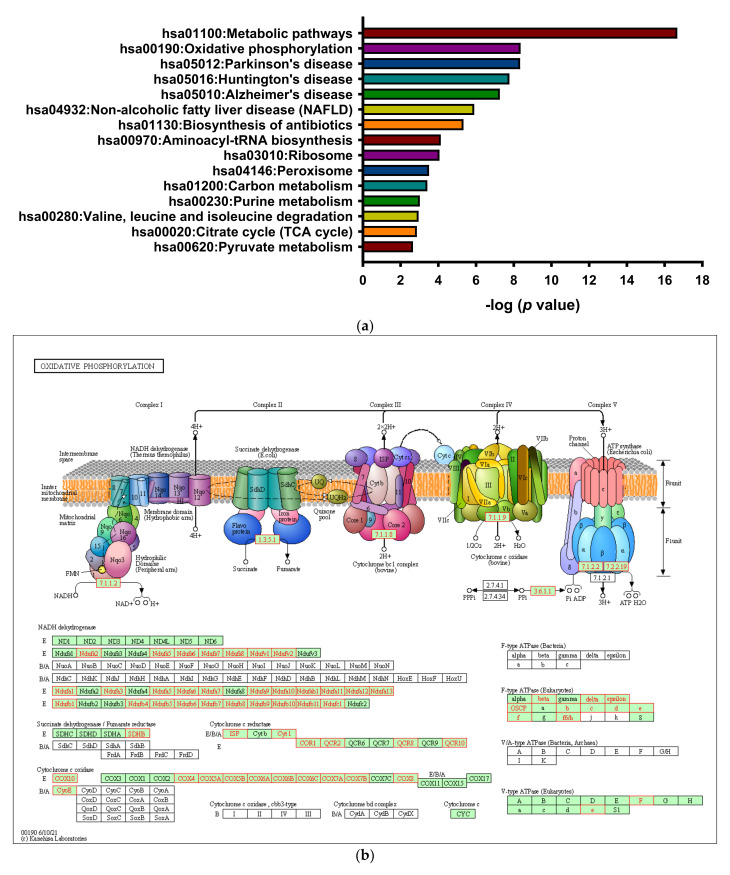
8-OHD-downregulated genes involved in oxidative phosphorylation. (**a**) The top 15 KEGG pathways associated with genes commonly downregulated by both 50 and 100 μM 8-OHD were analyzed with DAVID functional annotation cluster tool. Oxidative phosphorylation was the second most significantly enriched pathway. (**b**) KEGG “Oxidative Phosphorylation” pathway (hsa00190) associated with 8-OHD-downregulated genes. E: eukaryote; B: bacteria; A: archaea. Downregulated genes obtained from the microarray are marked in red. (**c**) Heatmap of expression patterns of some oxidative phosphorylation-associated genes in response to 8-OHD. H5 and H1 represent 50 and 100 μM 8-OHD, respectively.

**Figure 7 biomedicines-09-01907-f007:**
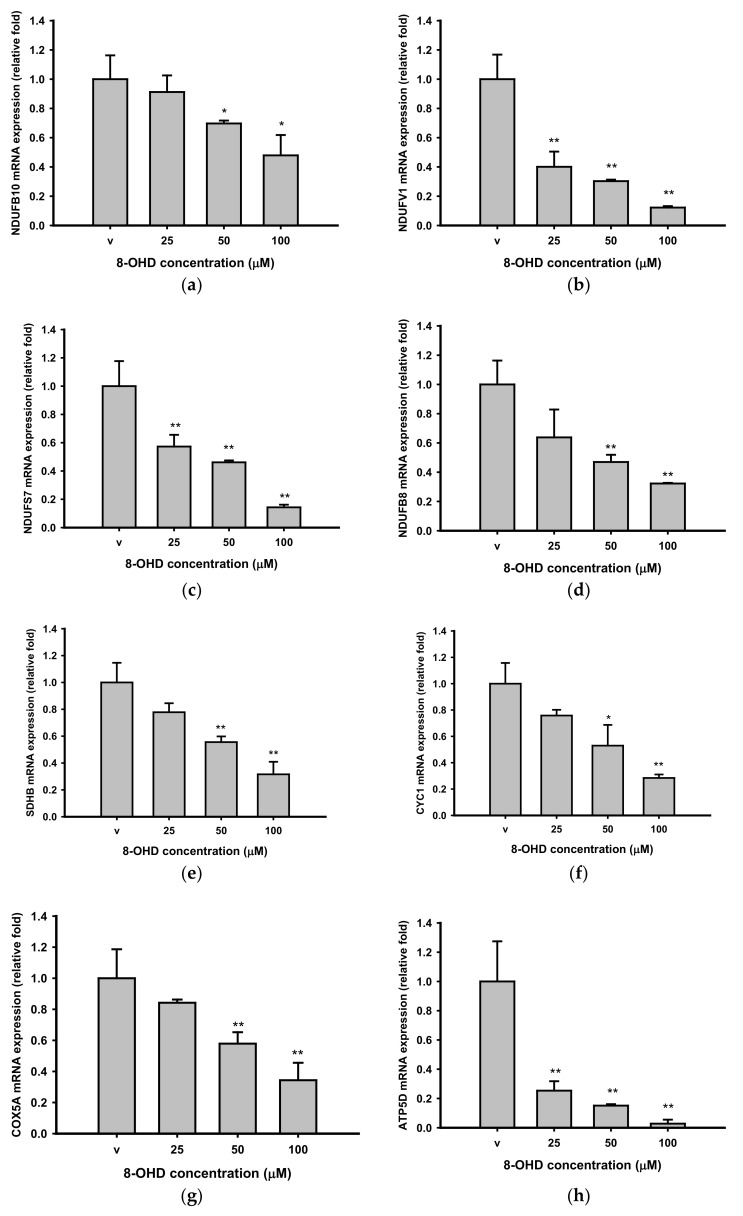
Reconfirmation of downregulation of oxidative phosphorylation genes by RT-qPCR. * *p* < 0.05 and ** *p* < 0.01 indicate significant differences compared to vehicle-treated cells.

**Figure 8 biomedicines-09-01907-f008:**
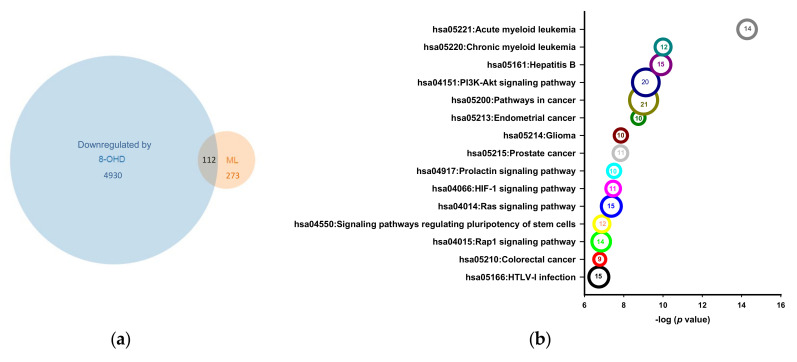
KEGG pathways associated with both 8-OHD-downregulation and myeloid leukemia (ML). (**a**) Venn diagrams demonstrating 112 putative ML targets downregulated by 8-OHD. (**b**) Bubble chart indicates the top 15 enriched KEGG pathways by 112 targets. Gene counts are indicated by the bubble size and number inside. (**c**) KEGG “Chronic myeloid leukemia pathway” (hsa05220) associated with 8-OHD-downregulated and ML genes. (**d**) KEGG “PI3K-Akt signaling pathway” (hsa04151) associated with 8-OHD-downregulated and ML genes. Green box indicates that map objects exist and are linked to corresponding entries. Downregulated genes obtained from microarray are indicated by red star signs.

**Figure 9 biomedicines-09-01907-f009:**
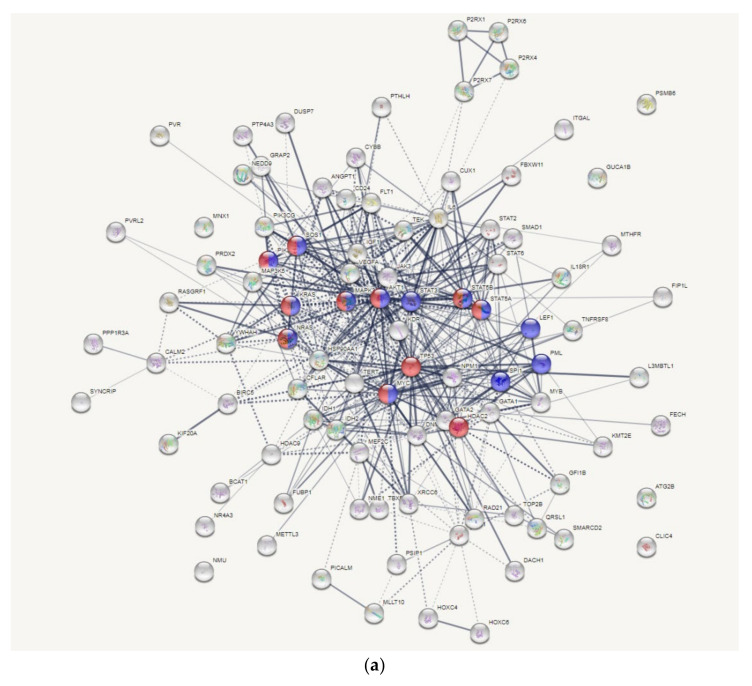
Predicted protein-protein interactions (PPIs) for 8-OHD-downregulated targets associated with myeloid leukemia (ML). (**a**) The PPI network of 112 targets. Corresponding genes were uploaded to query the STRING interaction database. Only the ‘‘Experiments’’, ‘‘Databases’’, and “Textmining” source options were selected, and the minimum interaction score was set to 0.4. The line thickness indicates the strength of data support. The solid and the dotted lines respectively indicate connections within the same and different clusters. Those proteins involved in the KEGG acute ML (AML) pathway and chronic ML (CML) pathway are respectively shown in blue and red. (**b**) Dose- and time-dependent inhibitory effects of 8-OHD on the phosphorylation of Akt (Ser473) in K562 cells. (**c**) Effects of 8-OHD on the nuclear c-Myc level. Lamin A/C was used as the loading control.

**Table 1 biomedicines-09-01907-t001:** Primary antibodies used for Western blotting.

Antibody	Company	Catalog No.
α-Tubulin	Sigma-Aldrich	T6199
JAK2	Cell Signaling	3230
STAT3	Cell Signaling	9132
p-STAT3	Cell Signaling	9134
AKT	Cell Signaling	9272
p-AKT	Cell Signaling	9271
c-Myc	Cell Signaling	5605
Lamin A/C	Genetex	GTX101127

**Table 2 biomedicines-09-01907-t002:** The primer pairs used in real-time PCR.

Gene		Primer Sequence	Amplicon (bp)
*GADPH*	F	CATGAGAAGTATGACAACAGCCT	113
	R	AGTCCTTCCACGATACCAAAGT	
*MMP11*	F	CCGCAACCGACAGAAGAGG	145
	R	ATCGCTCCATACCTTTAGGGC	
*MMP14*	F	GGCTACAGCAATATGGCTACC	83
	R	GATGGCCGCTGAGAGTGAC	
*MMP15*	F	AGGTCCATGCCGAGAACTG	157
	R	GTCTCTTCGTCGAGCACACC	
*VEGFA*	F	AGGGCAGAATCATCACGAAGT	75
	R	AGGGTCTCGATTGGATGGCA	
*SOS1*	F	GAGTGAATCTGCATGTCGGTT	177
	R	CTCTCATGTTTGGCTCCTACAC	
*ATP5D*	F	TCCCACGCAGGTGTTCTTC	178
	R	GGAACCGCTGCTCACAAAGT	
*CYC1*	F	CCAGGGAAGCTGTTCGACTAT	80
	R	GGCAATGCTCCGTTGTTGG	
*NDUFB8*	F	CCGCCAAGAAGTATAATATGCGT	204
	R	TATCCACACGGTTCCTGTTGT	
*NDUFS7*	F	CTTCGCAAGGTCTACGACCAG	90
	R	GGAATAGTGGTAGTAGCCTCCTC	
*NDUFV1*	F	AGGCCCAAGTATCTGGTGGT	85
	R	TGTGAGGATCATGGCGTAAGA	
*COX5A*	F	ATCCAGTCAGTTCGCTGCTAT	102
	R	CCAGGCATCTATATCTGGCTTG	
*NDUFB10*	F	AAAGCGTTCGACCTCATCGT	178
	R	TCTTCCACTGCATTTCGGCT	
*SDHB*	F	GTGGCCCCATGGTATTGGAT	142
	R	CGGGTGCAAGCTAGAGTGTT	

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
