# Peer review of "8-Hydroxydaidzein Downregulates JAK/STAT, MMP, Oxidative Phosphorylation, and PI3K/AKT Pathways in K562 Cells"

_biomedicines, 2021, doi:10.3390/biomedicines9121907_

Round 1

Reviewer 1 Report

K562 cells were treated with 8-OHD (50 and 100 μM) for 48 h, and differentially expressed genes (DEGs) were identified. The authors investigate the pathways and networks associated with the 1910 8-OHD-downregulated genes found in this study. Bioinformatic data were confirmed by experimental assays.

Bioinformatic analysis found several pathways affected by 8-OHD, mainly JAK2/STAT, PI3K/AKT, metalloproteases, oxidative phosphorylation, c-MYC and AKT.

The study shows the inhibitory effects of high doses of 8-OHD on these pathways previously reported as central both in CML and other cancers, thus revealing the molecular basis of the anti-cancer effects of the isoflavone.

The study is interesting and well performed, however the quality of figures is low, due to many panels partially cut. Figures 4,5,6,7,8 must be corrected.

Moreover, I wonder why several experimental data were obtained after stimulation only with low doses of 8-OHD (25 and 50 uM) and not with the 100 uM concentration. The discrepancy should be amended or justified.

In addition, I suggest a discussion about the impact of different doses in relation with ROS levels, because at 25 uM 8-OHD is reportedly a ROS scavenger in microglia and macrophages whereas 100 uM is a ROS producing dosage. Is this dual effect known in K562 or other leukemia cells? Are the effects observed in experimental data possibly due to ROS modulation?

Author Response

Reviewer 1

K562 cells were treated with 8-OHD (50 and 100 μM) for 48 h, and differentially expressed genes (DEGs) were identified. The authors investigate the pathways and networks associated with the 1910 8-OHD-downregulated genes found in this study. Bioinformatic data were confirmed by experimental assays.

Bioinformatic analysis found several pathways affected by 8-OHD, mainly JAK2/STAT, PI3K/AKT, metalloproteases, oxidative phosphorylation, c-MYC and AKT.

The study shows the inhibitory effects of high doses of 8-OHD on these pathways previously reported as central both in CML and other cancers, thus revealing the molecular basis of the anti-cancer effects of the isoflavone.

  1. The study is interesting and well performed, however the quality of figures is low, due to many panels partially cut. Figures4,5,6,7,8 must be corrected.

Response: We found that page layout of submitted manuscript was altered by editor (or someone else) so that several figures became truncated. We reset the format so that all of the original figures can be clearly seen in this revised manuscript.

  1. Moreover, I wonder why several experimental data were obtained after stimulation only with low doses of 8-OHD (25 and50 uM) and not with the 100 uM concentration. The discrepancy should be amended or justified.

Response: As mentioned in question 1, our figures were truncated and the high dose data were mistakenly deleted. In fact, all of the original experimental data were with 25, 50 and 100 mM concentrations. We have shown all the data in full in this revised manuscript.

  1. In addition, I suggest a discussion about the impact of different doses in relation with ROS levels, because at 25 uM 8-OHD is reportedly a ROS scavenger in microglia and macrophages whereas 100 uM is a ROS producing dosage. Is this dual effect known in K562 or other leukemia cells? Are the effects observed in experimental data possibly due to ROS modulation?

Response: Thank reviewer very much for this valuable suggestion. We have added “Similar to lots of natural compounds, 8-OHD exerts hormetic effect in resistance to oxidative stress [40]. It scavenges free radicals at low doses; while produces reactive oxygen species (ROS) at high doses [32,33,37,39,41,42]. It may play a dual roles, antioxidant and pro-oxidant, in cancer prevention and treatment.” in “Introduction” section.

Reviewer 2 Report

Thank you for the opportunity to read this interesting manuscript. The authors should discuss in this article the involvement of Blc-family proteins and describe if 8- OHD is able to modulate them since they are strictly correlated to mitochondrial pathways.

Round 2

Reviewer 1 Report

The article has been revised as suggested and it is now ready for publication.